# High-Strength Regenerated Cellulose Fiber Reinforced with Cellulose Nanofibril and Nanosilica

**DOI:** 10.3390/nano11102664

**Published:** 2021-10-11

**Authors:** Yu Xue, Letian Qi, Zhaoyun Lin, Guihua Yang, Ming He, Jiachuan Chen

**Affiliations:** 1School of Chemistry and Chemical Engineering, University of Jinan, Jinan 250022, China; xxyy0707@163.com; 2State Key Laboratory of Biobased Material and Green Papermaking, Qilu University of Technology, Shandong Academy of Sciences, Jinan 250353, China; lqi01@qlu.edu.cn (L.Q.); linzhaoyun@qlu.edu.cn (Z.L.); chenjc@qlu.edu.cn (J.C.)

**Keywords:** dissolving pulp, cellulose nanofibril, nano silicon dioxide, high strength

## Abstract

In this study, a novel type of high-strength regenerated cellulose composite fiber reinforced with cellulose nanofibrils (CNFs) and nanosilica (nano-SiO_2_) was prepared. Adding 1% CNF and 1% nano-SiO_2_ to pulp/AMIMCl improved the tensile strength of the composite cellulose by 47.46%. The surface of the regenerated fiber exhibited a scaly structure with pores, which could be reduced by adding CNF and nano-SiO_2_, resulting in the enhancement of physical strength of regenerated fibers. The cellulose/AMIMCl mixture with or without the addition of nanomaterials performed as shear thinning fluids, also known as “pseudoplastic” fluids. Increasing the temperature lowered the viscosity. The yield stress and viscosity sequences were as follows: RCF-CNF^2^ > RCF-CNF^2^-SiO_2_^2^ > RCF-SiO_2_^2^ > RCF > RCF-CNF^1^-SiO_2_^1^. Under the same oscillation frequency, G’ and G” decreased with the increase of temperature, which indicated a reduction in viscoelasticity. A preferred cellulose/AMIMCl mixture was obtained with the addition of 1% CNF and 1% nano-SiO_2_, by which the viscosity and shear stress of the adhesive were significantly reduced at 80 °C.

## 1. Introduction

Cellulose is a natural and abundant macromolecule that forms with D-glucopyranose ring units in the ^4^C_1_-chair configuration, which is linked by *β*-1, 4-glycosidic bonds that result in an alternate turning of the cellulose chain axis by 180° [1]. Complex hydrogen-bond (H-bond) networks are identified in cellulose macromolecules due to the presence of free hydroxy groups. Thereby, as the complex structure of cellulose makes it difficult to dissolve in water and common organic solvents, there are many restrictions on its use, as well as on its further development [2,3]. Despite the stiff and insoluble nature, the cellulose-based materials have many merits, including being lightweight [4], air permeable [5], etc. As being obtained from plants, they are commonly of low cost, naturally degradable and sustainable [1,6]. Therefore, the dissolution of cellulose is a key step in the production of cellulose-based materials [6]. In the early industrial era, copper ammonia and carbon disulfide methods were commonly used to produce cellulose fiber and film [7]. N-methylmorpholine (NMMO) is currently used in the production of Tencel [8,9]. However, these methods are suffering with severe chemical modification of cellulose, large solvent lost and environmental problems.

The discovery and rapid development of ionic liquids (ILs) reveals their unique solubility and H-bonding ability. Thereby, the application of ILs in dissolving cellulose have been extensively studied [10,11,12,13], in order to break the complex H-bond network of cellulose. Swatloski et al. first reported that room-temperature ILs could be used for soluble, non-derived cellulose [14]. Zhang et al. found that cellulose was completely soluble in 1-allyl-3-methylimidazolium chloride (AMIMCl) and 1-butyl-3-methylimidazolium chloride (BMIMCl), which could be used for producing cellulose film and fiber [7,15,16]. ILs such as 1-ethyl-3-methylimidazolium acetate (EMIMAc) have also been reported to prepare cellulose fibers and thin films [17,18]. The dissolved cellulose solution can be spun into regenerated cellulose fibers. Its tensile strength generally lies within the range of 1.6~5.34 cN/dtex, which is comparable to Lyocell [19]. In contrast, the present physical strength of ILs-regenerated fibers failed to meet the standard of high-strength materials, such as industrial textile, rope, triangle belt and so on.

Addition of nanomaterials, even with a small dosage, have been reported to significantly improve the physical strength of the composite materials [20]. Nanocellulose added to composite cellulose paper has been reported to significantly improve its mechanical strength [21]. Nanosilica particles are thoroughly and uniformly dispersed into resin materials, comprehensively enhancing their smoothness and aging resistance [22,23,24]. Song et al. found microcrystalline cellulose and nano-SiO_2_ composite fibers had good tensile strength and thermal stability [23]. Lee et al. used a wet-laid sheet-forming process to produce nanocomposites named CNF/PA6, improving the tensile strengths to 16.4 MPa [25]. He et al. added SiO_2_ nanoparticles into cellulose hydrogels to improve the transmittance and mechanical strength [26]. However, most of the literature research studies mainly focus on adding nanomaterials into polymer film and gel system [27,28]. There are only limited studies on the preparation of regenerated cellulose fibers by mixing nanomaterials and pulp in ionic liquid system. Herein, in this work, nanocellulose and nanosilica were added either individually or in combination into softwood-dissolving pulp/1-allyl-3-methylimidazolium chloride (AMIMCl) to improve the physical strength of regenerated fibers. The physical strength of regenerated cellulose fibers was examined, together with chemical, morphological and rheological characterization. The results can provide theoretical support for the preparation of high-strength cellulose fiber.

## 2. Materials and Methods

### 2.1. Materials

The bleached softwood dissolving pulp (DP~548; the content of alpha-cellulose was 88.74%.) was obtained from a pulp mill in Jining, China. Nanosilica (BET300 ± 50 m^2^/g, MW 60.08 g/mol, particle size 15 nm) was purchased from Macklin Inc., Shanghai, China. Cellulose nanofibril was obtained from cotton with a diameter of 4–10 nm, length of 1–3 μm, and carboxylic group content of 2.5 mmoL/g, which was purchased from Guilin Qihong Technology Co., Ltd. The 1-allyl-3-methylimidazolium chloride (AMIMCl) IL used in this study was synthesized according to the procedure described in the literature [7]. A successful synthesis was confirmed by nuclear magnetic resonance spectroscopy (AVANCEII400, Bruke Inc., Berlin, Germany), and the water content (0.51%) was verified by a Karl Fischer moisture meter(AKF-1, Mettler Toledo International Co., LTD, Zurich, Switzerland). Methyl imidazole and 3-allyl chloride were purchased from Macklin Inc., Shanghai, China. All of the other chemicals were analytical grade and used without further purification.

### 2.2. Dissolution and Regeneration Processing of Cellulose

The dissolution process consisted of adding cellulose nanofibers and nanosilica particles into the IL and heating to 70 °C. The dosages of CNF and nano silicon dioxide are shown in Table 1. Dissolved pulp (10 wt% in AMIMCl) was then added into the IL, until the pulp dissolved completely, to generate the cellulose viscose. Then we put the cellulose viscose into a single screw extruder with a single-hole extruder head, and the aperture of extruder head was 0.8 mm. The length–diameter ratio of extruder screw is 23:1. The speed is 50 rad per minute, the amount of viscose extrusion is 300 g/h. All of the fibers used the dry-jet wet-spinning process with a 1.5 cm air-gap and deionized water as a coagulation bath. The fiber was first soaked in deionized water for 24 h, washed with deionized water at 80 °C for 5 min, and then washed with deionized water to remove AMIMCl until the content of AMIMCl in the regenerated fiber was no more than 0.3 wt%. Then, the fibers were dried with hot air at 105 °C for 30 min. Finally, the regenerated cellulose fibers were placed under 23 °C and 50% relative humidity (RH) for 24 h. The fiber without CNF and nano-SiO_2_ was named RCF. The fiber with 2% CNF added was named RCF-CNF^2^. The fiber with 2% nano-SiO_2_ added was named RCF-SiO_2_^2^. The fiber with 1% CNF and 2% nano-SiO_2_ added was named RCF-CNF^1^-SiO_2_^1^. The fiber with 2% CNF and 2% nano-SiO_2_ added was named RCF-CNF^2^-SiO_2_^2^.

### 2.3. Characterization of Regenerated Fibers

#### 2.3.1. Tensile Properties of Fibers

The mechanical properties of the regenerated cellulose fibers were analyzed by using a texture analyzer (stable microsystems PL/CEL5, Stable Micro Systems Inc., London, UK), at 23 °C and 50% RH, with a load cell of 5000 g. The initial tensile distance was kept at a constant 20 mm, and the rate of the tensile test was constant at 10 mm/min. The diameter of each fiber sample was measured by using a Vernier caliper for 10 different places per 50 mm. The average diameter was used in the calculations. Each sample was tested at least 10 times.

#### 2.3.2. Contact Angle Test

Since the contact angle of the fiber surface was difficult to measure, it was performed by using cellulose films formed with the viscose extruded from the spinneret head. A water contact angle test was taken on an OCA50 (Dataphysics, Filderstadt, Germany), using 1 µL of liquid.

#### 2.3.3. X-Ray Diffraction (XRD) Analysis

The XRD data were collected on a model D8 ADVANCE diffractometer (Bruker AXS Co., Karlsruhe, Germany) in an angular region 2*θ* ranging between 5 and 60°, with a scanning speed of 5°/min. The fibers were tightly wound 20 times on the test sheet, covering the test sheet surface completely, and then put into the instrument for testing. Crystallinity was calculated through the diffraction patterns. The calculation method was as follows:Crystallinity Xc = (*I_101_*−*I_am_*)/*I_101_* × 100%(1)
where *I_101_* was the (101) surface diffraction intensity, and *I_am_* was the diffraction intensity of the amorphous region. For cellulose I, *I_am_* was the diffraction intensity of 2*θ* at 18.0°. For cellulose II, *I_am_* was the diffraction intensity of 2*θ* at 15.0° [29].

#### 2.3.4. Fourier-Transform Infrared (FTIR) Spectroscopy

The FTIR spectra of the regenerated cellulose fibers were recorded by using an ALPHA FTIR spectrophotometer with an attenuated total reflectance (ATR) (Bruker Corporation, Billerica, MA, USA). All of the spectra were obtained from 32 scans with a resolution of 4 cm^−1^ and absorption mode, using a wavelength range from 500 to 4000 cm^−1^. At least three repetitions per sample were conducted.

#### 2.3.5. Scanning Electron microscopy (SEM) and Energy-Dispersive X-ray Spectroscopy (EDS)

Images of the regeneration cellulose fiber samples were taken on a regulus8220 (Hitachi Ltd., Tokyo, Japan) microscope operating at 5 kV. The fracture surface of the fibers was coated with a platinum layer (with a thickness of approximately 20 nm) before observation. Moreover, the surface element content of the fibers was obtained via EDS on a regulus8220 (Hitachi Ltd., Tokyo, Japan) at 10 kV.

#### 2.3.6. Thermal Analysis

Thermographic analysis (TGA) of the fibers was performed on a simultaneous thermal analyzer (TA TGAQ50, TA Instruments, New Castle, DE, USA). A total of 30~40 mg of the sample was placed in an aluminum oxide pan and heated from 35 to 800 °C, at a rate of 10 °C/min, in a nitrogen atmosphere, with a flush rate of 40 mL/min.

#### 2.3.7. Rheological Measurements

Steady and dynamic rheology experiments were carried out on an TA ARES-G2 rotated rheometer (TA Instruments, New Castle, DE, USA)). Parallel plates (20 mm in diameter) were used. The chosen gap was 1 mm for all of the measurements. The strain amplitude values were checked to ensure that the measurement deviation did not exceed 5%, which was in the linear viscoelastic regime. The shear rate ranged from 0.1 to 100 s^−1^, and the sweep of the angular frequency (w) was 6.28 rad/s. Dynamic viscoelastic functions, such as the shear storage modulus (elasticity modulus) (G’) and loss modulus (viscous modulus) (G”) as a function of time, angular frequency, and temperature, were also measured.

## 3. Results and Discussion

### 3.1. Tensile Properties

It was reported that the addition of nanomaterials could increase the strength of polymers [24,30], similar results were found in this work. As can be seen from Table 2, the tensile strength of RCF is 155.82 MPa, and the elongation at break is 8.73%. After adding CNF and nano-SiO_2_, the tensile strength of all the regenerated cellulose fibers was improved, as the tensile strength of RCF-CNF^2^ is 174.15 MPa, which is increased by 11.76% in comparison with RCF. Its broken number in spinning is 5, and the elongation at break is 7.43%, which is decreased by 14.89%. Similarly, the tensile strength of RCF-SiO_2_^2^ is 174.18 MPa, which is increased by 11.76% than that of RCF. Its elongation at break is 7.65%, which is decreased by 12.37%. Therefore, the tensile strength of RCF-CNF^2^ and RCF-SiO_2_^2^ increase by the same extent.

The enhancement of physical strength implied that the nanomaterials added closely interacted with regenerated cellulose, instead of just attaching on the fiber surface. Both nanocellulose and softwood-dissolving pulp can be dissolved in AMIMCl because of the strong coulombic and H-bond interactions within the IL system [31]. Interacted by the IL anions and cations, the intermolecular and intramolecular hydrogen bonds of cellulose chains are broken, which results in the cellulose dissolution and formation of homogeneous solution [15,32]. The length of softwood dissolving pulp is normally 0.5~0.6 mm, whereas the typical length of CNF is 0.001~0.003 mm. Since CNF has a short molecular chain, large specific surface area and free active carboxyl groups, during the regeneration process, it can interact closely with cellulose fiber chains through H-bonding, resulting in the enhancement of their physical strength.

In terms of nano-SiO_2_, as an inorganic compound, it was added into cellulose viscose system to form an organic–inorganic hybrid viscose system. As nano-SiO_2_ is difficult to dissolve in AMIMCl, it was only dispersed as suspensions. The excellent specific surface area allowed nano-SiO_2_ to have strong interfacial adsorption through van der Waals interactions with cellulose [23]. Nano-SiO_2_ can also bind with the small amount of water presented in cellulose/ILs system, where nano-SiO_2_, cellulose, and water are tightly bonded to form a complex network structure [26,33].

According to the literature, it was expected that increasing dosage of nanomaterials would lead to a further enhancement in strength [23]. However, the fiber strength of RCF-CNF^2-^SiO_2_^2^ was the same as RCF-CNF^2^ and RCF-SiO_2_^2^. The tensile strength of RCF-CNF^2-^SiO_2_^2^ was 173.42 MPa, which was increased by 11.29% than that of RCF. Its elongation at break of RCF-CNF^2-^SiO_2_^2^ was 6.33%, which was decreased by 27.49%. Interestingly, it was found that the addition of 1% CNF and 1% nano-SiO_2_ resulted in a significant increase in the fiber strength, and the elongation at break also dramatically increased. It should be noticed that, in all other testing samples, the elongation at break were all decreased compared with the RCF sample. The tensile strength of RCF-CNF^1-^SiO_2_^1^ was 229.78 MPa, an increase of 47.46%, which was the strongest of all the fibers; and the elongation at break of RCF-CNF^1^-SiO_2_^1^ was 10.01%, an increase of 14.66%. This indicated that RCF-CNF^1^-SiO_2_^1^ may make the viscose system and regeneration process different from other conditions, in order to provide a tightly interacted composite fiber.

### 3.2. Water Contact Angle

Nano-SiO_2_ is a common surface hydrophobic agent that can give fibers a particular hydrophobic property when coated on their surface [24]. Cellulose has excellent hydrophilicity due to the presence of a large number of hydroxyl groups in its structure [33,34,35]. Therefore, the addition of these nanomaterials should alter the hydrophilicity of the regenerated fibers. Since the contact angle of the fiber surface was difficult to measure, it was performed by using cellulose films formed with the viscose extruded from the spinneret head.

Results in Figure 1 show the change of contact angle with the addition of nanomaterials. The cellulose film without nanomaterials was 51.85°, whereas the contact angles of the cellulose film with RCF-CNF^2^, RCF-SiO_2_^2^, RCF-CNF^1^-SiO_2_^1^, and RCF-CNF^2^-SiO_2_^2^ were 62.1, 69.15, 67.05, and 78.9°, respectively. Therefore, the hydrophily of all the films with nanomaterials added was reduced. It was assumed that the addition of nanocellulose interacted with regenerated cellulose, making the surface smooth and dense, and thus resulting in greater contact. Meanwhile, as a hydrophobic agent, the presence of nano-SiO_2_ on the surface reduced hydrophilicity and increased the contact angle [33].

### 3.3. XRD Analysis

Figure 2 and Table 3 showed the XRD patterns and crystallinity results of raw materials and fibers with or without nanomaterials added. As can be seen from the XRD images, the configuration of conifer pulp was of cellulose I type crystallographic form, and the regenerated cellulose fibers made were all of cellulose II form. The bleached softwood dissolving pulp was dissolved in AMIMCl, and the hydrogen bonds between the cellulose molecules were broken. The cellulose solution entered the coagulant through the extruder head, after which the AMIMCl in the viscose dissolved in the water, and hydrogen bonds between the cellulose molecules reformed. Glucose units were bonded together by a secondary helix structure during the formation process to produce cellulose II type of crystallography. Meanwhile, the results showed that the addition of CNF and nano-SiO_2_ did not affect the crystal structure of cellulose in the regeneration process, which also illustrated that the CNF added was also dissolved in AMIMCl and regenerated into cellulose II [1,15,36].

During the spinning process, the cellulose macromolecular chains were oriented by high shear and subsequent stretching in the air gap, which induced crystallization during drying. It is known that the tenacity of regenerated fibers mainly depends on the orientation of their amorphous region, while the modulus and crystallinity are related to crystal orientation [19]. Thus, the crystallinity and orientation determine the mechanical properties of the regenerated fibers.

The crystallinity of the pulp was 50.88%, and the crystallinity of RCF was 62.76%, with an increase of 10%. This was because the hydrogen bond length of cellulose II was shorter than that of cellulose I. As a result, the cellulose molecular chains were more tightly stacked, and their structure was more thermodynamically stable [37]. The crystallinity of RCF-CNF^2^ was 50.11%, which was about 20.15% lower than that of RCF. The crystallinity of RCF-SiO_2_^2^ was 47.37%, which was the lowest among the samples. The crystallinity of RCF-CNF^2-^SiO_2_^2^ was 47.75%, which was basically the same as that of RCF-SiO_2_^2^. However, the crystallinity of RCF-CNF^1-^SiO_2_^1^ sample was significantly elevated to 55.15%, which was higher than those of other nanomaterials added samples. However, this value was still lower than that of RCF sample. Therefore, it could be concluded that the crystallinity of the fibers decreased with the addition of CNF and nano-SiO_2_.

It is generally believed that fibers with higher crystallinity usually present stronger physical strength, which could be attributed to the rigid and stiff nature of crystalline regions [19,23,24]. However, the results presented in this work, where samples with the addition of nanomaterials showed lower crystallinity but higher tensile strength. Therefore, the characterization over their chemical interactions, morphology and rheological behaver would be crucial to understanding the mechanism of strength enhancement.

### 3.4. Fourier-Transform Infrared (FTIR) Spectroscopy Analysis

Considering the fact that all testing samples contained ~98% of cellulose, the FTIR spectra in Figure 3 shows similar trend, indicating that the structure and composition of the fibers were similar. The peak in the range 2900~2800 cm^−1^ was related to stretching of C–H from methyl groups. The peak in the range 1400~1150 cm^−1^ corresponded to the deformations of C=OH. The peak in the range 1135~1180 cm^−1^ corresponded to the deformations of the C–OH, C–H, and C–O–C groups of cellulose. The bond at 1100 cm^−1^ was considered to be the characteristic peak of C–C. The 887 cm^−1^ peak was characteristic of the *β*-glycoside bond between the glucose units. The peak in the range 700~450 cm^−1^ corresponded to the vibration bending of O–H. The 804 cm^−1^ peak corresponded to the symmetric stretching of Si–O–Si [23,24,38,39,40,41,42], indicating the presence of SiO_2_ on the fiber surface.

However, minor difference in the intensity of the peaks could be identified. The band in the range 3600~3200 cm^−1^ is corresponded to the vibrational modes of the hydroxyl group in the cellulose molecules, particularly the band at 3500 cm^−1^ represented the stretching characteristic of OH groups, and the bond at 3246 cm^−1^ was the characteristic peak of hydrogen bonding. The changes in the OH groups’ related peaks indicate that the addition of CNF causes an increase in free OH groups, whereas the addition of nano-SiO_2_ results in a significant reduction of them. Similar trend was noticed at 1640~1630 cm^−1^, which was related to the water adsorption of the fibers. In comparison with the RCF sample, water adsorption in RCF-CNF^2^ increased, while RCF-SiO_2_^2^ was significantly reduced. This result may attribute to the hydrophobicity of the silica nanoparticles. However, it is interesting to note that the addition of CNF at the same dosage cannot overcome the reduction caused by nano-SiO_2_, where RCF-CNF^1^-SiO_2_^1^ and RCF-CNF^2^-SiO_2_^2^ samples present similar bands with that of RCF-CNF^2^.

Therefore, the infrared spectrum analysis shows that no derivatization reaction appears during cellulose regeneration. It has been reported that the cellulose dissolution and regeneration in AMIMCl is a physical process [7], which mainly involves the breaking and recombination of hydrogen bonds without chemical reactions. Thus, it is believed that the increase of fiber strength in this work was mainly caused by the rheology properties of viscose and the action of secondary bonds, including hydrogen bonding and interface adsorption [24].

### 3.5. SEM and EDS Analysis

The SEM images shows the surface morphology of the raw materials (Figure 4(a1,a2)) and regenerated fibers. As expected, the softwood dissolving pulp was a flat fiber with a layered fiber structure on its surface (Figure 4a) and some fine and filamentation fibers on the surface (Figure 4(a2)). After regeneration via dry spraying and wet spinning, the fiber surface presented as a smooth, filamentous structure (Figure 4b). Moreover, the fiber surface was a dense, scaly structure with pores and micro protrusions (Figure 4(b2)). The fiber surface with RCF-CNF^2^ added was relatively smooth (Figure 4(c1,c2)). The RCF-SiO_2_^2^ had some protuberance prominences on the surface (Figure 4(d1,d2)). The appearance of small particles may attribute to the aggregation of nano-SiO_2_ nanoparticles and the incompatibility of organic and inorganic materials. Meanwhile RCF-CNF^1-^SiO_2_^1^ had the densest and smoothest surface structure (Figure 4(e1,e2)), and the RCF-CNF^2-^SiO_2_^2^ was also relatively dense but with more prominences (Figure 4(f1,f2)). Adding CNF and nano-SiO_2_ simultaneously makes the material smooth and flat. The literature has reported similar results, wherein the addition of CNF and nano-SiO_2_ resulted in a smooth composite surface and could lead to an enhancement of mechanical properties [22,23,43].

It can be seen via EDS analysis that CNF and nano-SiO_2_ were retained in the fiber (Table 4), as the Si content of RCF and RCF-CNF^2^ was 0.15%, while the Si content of RCF-SiO_2_^2^, RCF-CNF^1^-SiO_2_^1^, and RCF-CNF^2^-SiO_2_^2^ was 0.56%, 0.49%, and 0.70%, respectively. The distributions of Si on the fibers’ surface were uniform (Figure 4g–l). However, the silica content on the surface was lower than the added amount. Therefore, it was speculated that most of the silicon atoms were embedded inside the cellulose fibers, where strong interactions between nano-SiO_2_ and cellulose macromolecules formed [23,44,45].

### 3.6. Thermal Analysis

Figure 5 shows that the regenerated cellulose fibers all exhibited typical features of cellulose degradation. The first degradation stage was from room temperature to about 150 °C and was mainly associated with the fiber dehydration and volatilization of some unstable substances. The second stage involved the decomposition of the amorphous region in the fiber structure, and part of the glucose group began to dehydrate. The temperature of this stage was generally between 150 and 250 °C. The fastest degradation rate occurred between 250 and 400 °C, during which the glycosidic bonds of cellulose broke and cellulose began to degrade [46,47,48].

The rapid degradation temperature of RCF-CNF^2^ was lower than the other four samples because the nanostructure of CNF made it easy to degrade, reducing the temperature of large-scale degradation of the fibers. The fastest degradation rate of RCF-SiO_2_^2^ was lower than RCF, and the thermal stability of the cellulose was slightly improved with the addition of 2% nano-SiO_2_. The distribution of silica can prevent the migration of small molecules in the fibers, so the decomposition of the cellulose macromolecular chain was delayed in the process of thermal decomposition [23,24].

From Table 5. it can be seen that final residual amount of fiber without nanomaterial was 22.58%. RCF-CNF^2^ started to degrade earlier than other samples, and the final residual amount was 20.39%. The fiber with added 2% CNF was completely degraded. The residual amount of the fiber with added 2% nano-SiO_2_ was 25.80%. The residual amount of RCF-CNF^1^-SiO_2_^1^ was 23.05%, and the residual amount of RCF-CNF^2-^SiO_2_^2^ was 24.93%. These results indicating that only minor loss of nano-SiO_2_ occurred in the regeneration process [23]. By taking the EDS results into consideration, this confirmed the previous hypothesis that most of the nano-SiO_2_ was embedded into the cellulose fibers during the regeneration process.

### 3.7. Rheological Properties of Cellulose/IL Solutions with Different Contents of Nanomaterial

In the process of viscose regeneration, the fluidity of viscose has a direct effect on the properties of regenerated cellulose fibers, and the viscosity is closely related to the phenomenon of filament breaking and filament doubling in the spinning process. To explore the reasons for the changes of fiber strength and crystallinity, the rheological properties of the viscose system during the extruding was tested. The viscose sample was obtained at the extruder head. The temperature of viscose at the extruder head was 80 °C and that of coagulation bath was 25 °C. Thus, the viscosity, stress, and viscoelasticity analysis of the viscose was carried out at 40, 60, and 80 °C.

From Figure 6, it can be seen that all testing viscose systems were shear-thinning system. The yield stress and viscosity sequence were as follows: RCF-CNF^2^ > RCF-CNF^2^-SiO_2_^2^ > RCF-SiO_2_^2^ > RCF > RCF-CNF^1^-SiO_2_^1^ viscose. For a low shear rate, the shear force produced a slight shearing orientation effect on the viscose, but the components in the system were still in a state of free distribution, and the shear viscosity was basically unchanged.

RCF-CNF^2^ had the largest initial viscosity of all the samples at 80 °C. This was mainly because CNF was identical to the wood fibers in terms of chemical structure, making it evenly dissolve and distribute in the viscose. Thus, the network structure of cellulose and ILs formed, and within the viscose, the CNF, pulp, and IL formed good crosslinking and hydrogen bonds to become a homogeneous mixing system characterized by a high yield stress value [2].

Comparatively, the viscosities of RCF-CNF^2^-SiO_2_^2^ and RCF-SiO_2_^2^ were lower than RCF-CNF^2^ viscose, but the viscosities of RCF-CNF^2^-SiO_2_^2^ and RCF-SiO_2_^2^ and the viscosity trend were of the same. This may be due to the nano-SiO_2_ reducing the generation of hydrogen bonds among the components [23], which were replaced by interfacial adsorption, thus lowering the overall viscosity. The viscosity of RCF-CNF^1^-SiO_2_^1^ was the lowest of all the samples, which meant that this system had less entanglement of macromolecules and force among components, or less network structures, better fluidity of the viscose, and more continuous viscose in the process of extruding. The force between each component was less affected by the pressure change of the spinneret, so it was difficult to break and strand, which was more conducive to the regeneration of cellulose and made the fiber surface smoother. This was consistent with the surface morphology characterized by SEM, which was also the reason for the higher fiber strength.

From the results shown in Figure 6, we can observe that, with the increase of shear rate, the apparent viscosity of several kinds of solutions decreased obviously to show shear thinning, which is typical of non-Newtonian fluids. The entangled macromolecules began to align in a directional manner, showing a decrease in apparent viscosity. When cellulose was completely dissolved in the IL, the van der Waals forces produced physical crosslinking between the molecules [49,50]. Hydrogen bonds existed between the cellulose macromolecules and macromolecules, and between cellulose the macromolecules and ions. The Brownian motion of each component in the viscose system made the physical crosslinking and hydrogen bonds exist in a dynamic equilibrium of constant destruction and regeneration, which made the viscose form a network structure. With the increase of shear force, the dynamic equilibrium was destroyed; the physical crosslinking and hydrogen bonds were not ready to recombine; and there appeared a relative slip and rearrangement between the components, resulting in a decrease in viscosity that eventually reached a constant value [51,52,53]. The shear rate had little influence on the viscosity of RCF-CNF^1^-SiO_2_^1^; that is to say, the state of the whole system was more stable, even during extrusion, and the tensile strength of the fiber was better.

Figure 7 shows the storage modulus G’ and loss modulus G” of the testing viscose. For the viscose system without nanomaterials, G’ was greater than G’’ at 40 °C, indicating that the elastic modulus was greater than the viscous modulus, which showed typical solid-like properties. The main forces of the system were based on the entanglement between cellulose molecules and the network structure between cellulose/AMIMCl. At temperatures of 60 and 80 °C, G’’ was greater than G’, indicating that the viscous modulus was greater than the elastic modulus, showing fluid properties, which also indicated that the properties of viscose were sensitive to the temperature. As the temperature changed, the viscoelasticity also changed. With the increase of temperature, the cellulose macromolecules were more active in AMIMCL, making the viscose appear fluid.

The viscoelasticity of viscose mainly depends on the properties of each component and their interactions. It is generally believed that the addition of nano-materials can change the viscoelasticity of viscose. For viscose with added CNF and nano-SiO_2_, when the temperature was constant, with the increase of frequency, the solution G’ and the solution G’’ both increased. Furthermore, values exhibited an intersection point, which is called the fiber solution gel point. With the increase of temperature, the intersection point moved toward the high-frequency area, where modulus transposition occurs, and the sample experienced structure reconstruction and an elasticity larger than the viscosity, which are typical characteristics of an entangled polymer solution. Under the same oscillation frequency, when the temperature increased, G’ and G” decreased, which indicated that the viscoelasticity decreased [54,55,56].

At 80 °C, G’ and G’’ of RCF-CNF^2^ were the largest among all of the samples, which indicated that the entanglement degree between the fiber molecules in the viscose system was higher, and the physical entanglement and hydrogen bonds among the cellulose macromolecules, CNF, and IL formed a tangled network structure that affected the relative motion and stretching of the molecules. Therefore, the viscose showed an obvious elasticity. The intersection point of G’ and G’’ experienced a minimal oscillation frequency, indicating that as adding CNF increased the cellulose content within the viscose system, the gel point of the whole system moved forward. Thus, in the regeneration process, solidification was easier. The G’ and G values of RCF-SiO_2_^2^, RCF-CNF^1^-SiO_2_^1^, and RCF-CNF_2_-SiO_2_^2^ were all lower than those of the blank samples, indicating that adding nano-SiO_2_ made the network structure in the system worse and reduced the generation of partial hydrogen bonds.

As shown in Table 6, for RCF-SiO_2_^2^ at a temperature of 80 °C, the intersection points between G’ and G’’ experienced an oscillation frequency of 102.78 rad/s, with a higher angular frequency of gelatinization. For RCF-CNF^2^-SiO_2_^2^ and RCF-CNF^1^-SiO_2_^1^, their G’ were similar to G’’, which indicated that adding equal amounts of CNF and nano-SiO_2_ in the system basically had the same effect on the entangling of macromolecules and changing the forces between the components. For RCF-CNF^1^-SiO_2_^1^, its intersection points of G’ and G’’ experienced an oscillation frequency of 90.28 rad/s. For RCF-CNF^2^-SiO_2_^2^, the intersection points of G’ and G’’ was at 100.79 rad/s. In the RCF-CNF1-SiO21 system, the degree of aggregation and entanglement among the components was greater than that of RCF-CNF^2^-SiO_2_^2^.

## 4. Conclusions

In this study, conifer pulp was used as a raw material to make regenerated fiber, an ionic liquid AMIMCl was used as a solvent for cellulose, and organic CNF fiber and inorganic nano-SiO_2_ were added to regenerate composite cellulose fibers via a dry-jet wet-spinning process. The mechanical properties of the composite cellulose can be improved by adding nanomaterials. Adding 1% CNF and 1% nano-SiO_2_ to the pulp/AMIMCl improved the mechanical properties of the composite cellulose by 47.46%. The regenerated fibers and nanocomposite fibers were typical type-II cellulose crystalline forms, and the crystallinity of the composite fibers decreased. The surface of the regenerated fiber exhibited a scaly structure with pores, and the pores could be reduced by adding 1% CNF and 1% nano-SiO_2_. The addition of nanomaterials changed the degradation rate of the regenerated fibers. From the perspective of the residual amount, the loss rate of nanomaterials in the preparation process was low. The viscosities of all five samples were pseudoplastic fluids with a yield value of stress and with shear thinning. The yield stress and viscosity sequences were as follows: RCF-CNF^2^ > RCF-CNF^2^-SiO_2_^2^ > RCF-SiO_2_^2^ > RCF > RCF-CNF^1^-SiO_2_^1^. Under the same oscillation frequency, when the temperature increased, the G’ and G” decreased, indicating a decrease in viscoelasticity.

## Figures and Tables

**Figure 1 nanomaterials-11-02664-f001:**
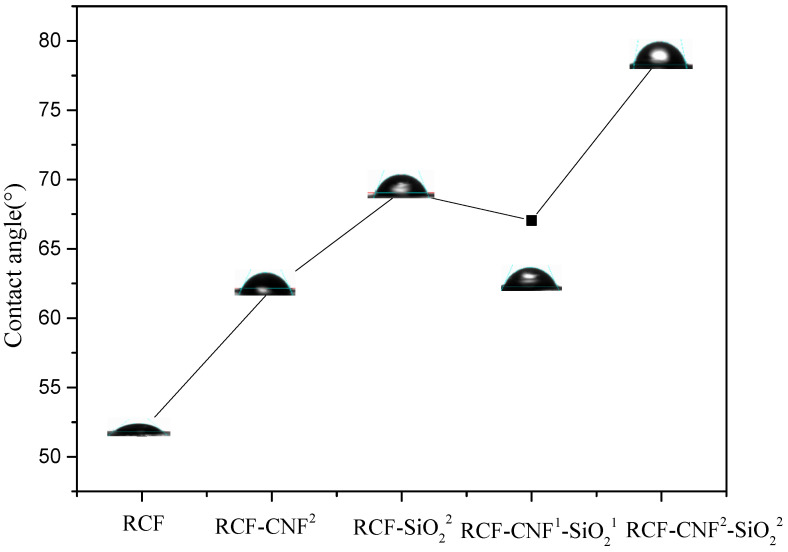
Contact angle of the regenerated cellulose films prepared with or without nanomaterials.

**Figure 2 nanomaterials-11-02664-f002:**
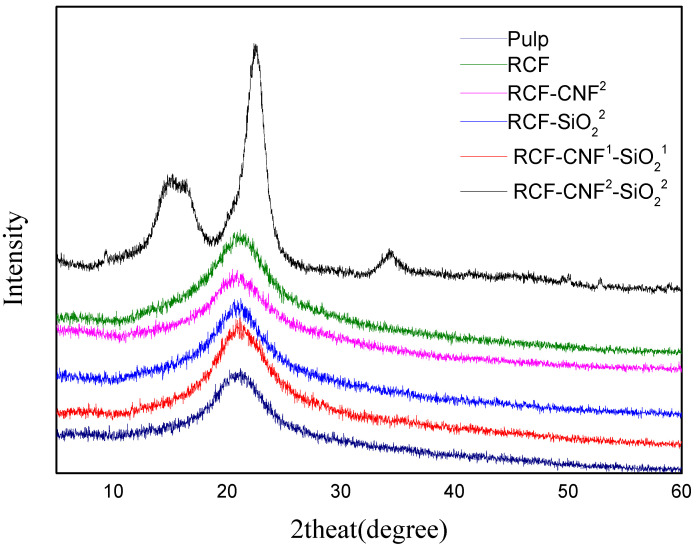
XRD analysis of the regenerated fibers with or without nanomaterials versus control cellulose pulp.

**Figure 3 nanomaterials-11-02664-f003:**
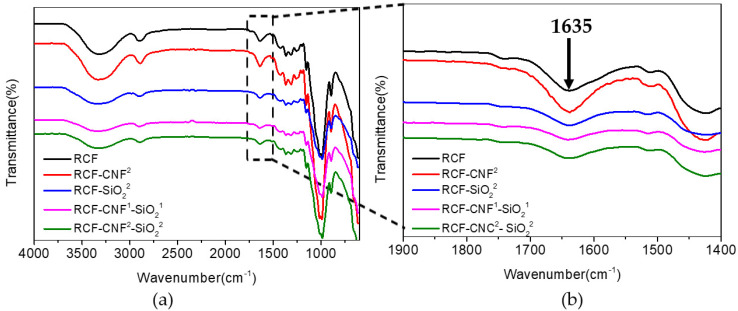
FTIR spectra of RCF, RCF-CNF^2^, RCF-SiO_2_^2^, RCF-CNF_1_-SiO_2_^1^, and RCF-CNF^2^-SiO_2_^2^: (**a**) Transmittance of fibers between the wavelength range from 4000 to 500 cm^−1^; (**b**) transmittance of fibers between the wavelength range from 1900 to 1400 cm^−1^.

**Figure 4 nanomaterials-11-02664-f004:**
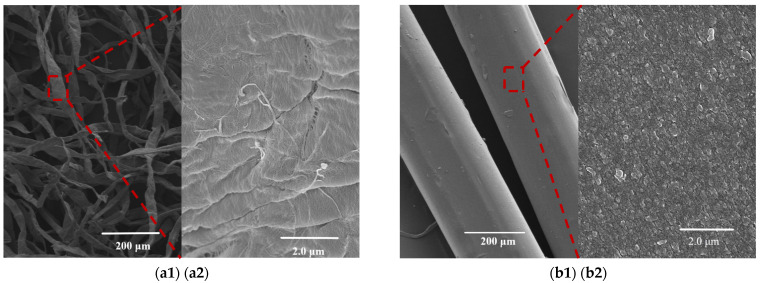
SEM and EDS images of the regenerated cellulose fibers: SEM images of pulp at ×100 (**a1**) and ×10,000 (**a2**), RCF at ×100 (**b1**) and ×10,000 (**b2**), RCF-CNF^2^ at ×100 (**c1**) and ×10,000 (**c2**), RCF-SiO_2_^2^ at ×100 (**d1**) and ×10,000 (**d2**), RCF-CNF^1^-SiO_2_^1^ at ×100 (**e1**) and ×10,000 (**e2**), RCF-CNF^2^-SiO_2_^2^ at×100 (**f1**) and ×10,000 (**f2**); EDS images of Pulp at ×2000 (**g**), RCF at ×1000 (**h**), RCF-CNF^2^ at ×1000 (**i**), RCF-SiO_2_^2^ at ×1000 (**j**), RCF-CNF^1^-SiO_2_^1^ at ×1000 (**k**), and RCF-CNF^2^-SiO_2_^2^ at ×1000 (**l**).

**Figure 5 nanomaterials-11-02664-f005:**
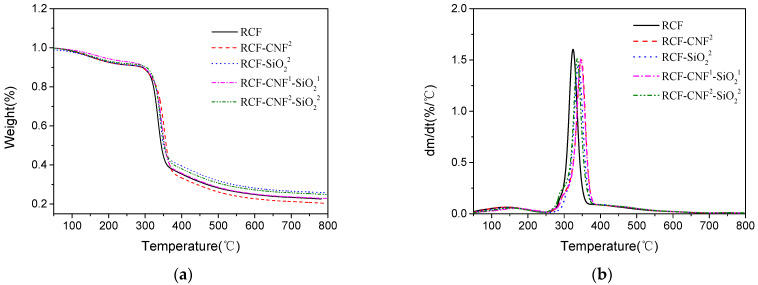
TGA (**a**) and DTA (**b**) curve of the regenerated cellulose fibers with or without nanomaterials.

**Figure 6 nanomaterials-11-02664-f006:**
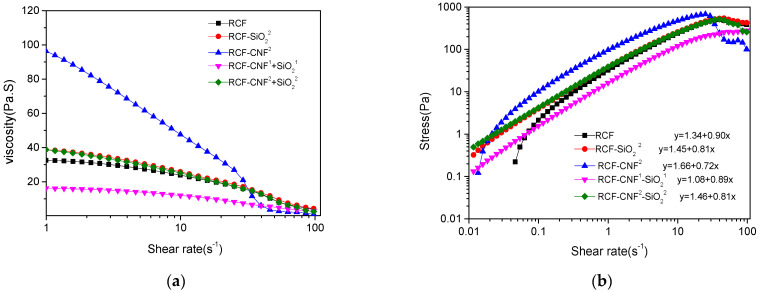
Viscosity (**a**) and shear stress (**b**) of cellulose viscose systems with or without nanomaterials at 80 °C.

**Figure 7 nanomaterials-11-02664-f007:**
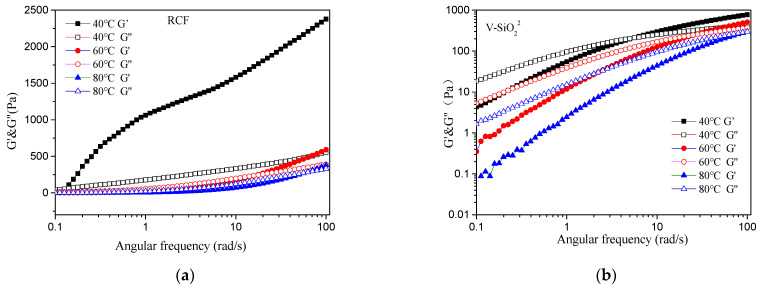
Influence of temperature and angular frequency on G’ and G” of RCF (**a**), RCF-CNF^2^ (**b**), RCF-SiO_2_^2^ (**c**), RCF-CNF^1^-SiO_2_^1^ (**d**), and RCF-CNF^2^-SiO_2_^2^ (**e**).

**Table 1 nanomaterials-11-02664-t001:** Dosage of CNF and nano-SiO_2_ in preparation of regenerated cellulose fibers.

Sample	CNF (wt% to Dissolving Pulp)	Nano-SiO_2_ (wt% to Dissolving Pulp)
RCF	/	/
RCF-CNF^2^	2%	/
RCF-SiO_2_^2^	/	2%
RCF-CNF^1^-SiO_2_^1^	1%	1%
RCF-CNF^2^-SiO_2_^2^	2%	2%

**Table 2 nanomaterials-11-02664-t002:** Tensile properties of the regenerated cellulose fibers prepared with or without nanomaterials.

Sample	Tensile Strength (MPa)	Elongation at Break (%)	Number of Broken in Spinning (/h)
RCF	155.82 ± 5.23	8.73 ± 0.13	1
RCF-CNF^2^	174.15 ± 2.36	7.43 ± 0.11	5
RCF-SiO_2_^2^	174.18 ± 1.26	7.65 ± 0.15	3
RCF-CNF^1^-SiO_2_^1^	229.78 ± 0.91	10.01 ± 0.15	2
RCF-CNF_2_-SiO_2_^2^	173.42 ± 3.35	6.33 ± 0.09	2

**Table 3 nanomaterials-11-02664-t003:** Crystallinity of the regenerated fibers with or without nanomaterials versus control cellulose pulp.

Sample	Crystallinity (%)
Pulp	50.88 ± 1.03%
RCF	62.76 ± 1.21%
RCF-CNF^2^	50.11 ± 0.96%
RCF-SiO_2_^2^	47.37 ± 1.54%
RCF-CNF^1-^SiO_2_^1^	55.15 ± 0.53%
RCF-CNF^2-^SiO_2_^2^	47.75 ± 0.75%

**Table 4 nanomaterials-11-02664-t004:** Relative element content of the regenerated fibers surface with or without nanomaterials and control cellulose pulp by EDS.

Sample	C (%)	O (%)	N (%)	Si (%)
Pulp	28.2	46.32	25.35	0.14
RCF	28.05	46.63	25.16	0.15
RCF-CNF^2^	27.48	47.51	24.85	0.15
RCF-SiO_2_^2^	27.42	47.35	24.67	0.56
RCF-CNF^1-^SiO_2_^1^	26.87	48.53	24.11	0.49
RCF-CNF^2-^SiO_2_^2^	27.51	47.23	24.56	0.70

**Table 5 nanomaterials-11-02664-t005:** Thermal properties of the regenerated cellulose fibers after different stages of treatment.

Sample	T_on_ (°C)	Mass Loss (%)	T_max_ (°C)	Mass Loss (%)	Residue at 800 °C (%)
RCF	307	12.28	329	27.16	22.48
RCF-CNF^2^	300	10.96	344	30.44	20.39
RCF-SiO_2_^2^	308	11.76	343	30.42	25.80
RCF-CNF^1^-SiO_2_^1^	302	10.96	344	36.25	22.85
RCF-CNF^2^-SiO_2_^2^	313	11.76	335	26.42	24.93

**Table 6 nanomaterials-11-02664-t006:** Modulus values and frequency at the crossover point of G’ and G” of cellulose viscose.

Sample	Temperature (°C)	Angular Frequency (Rad/s)	Modulus Values (Pa)
RCF	40	0.13	56.45
	60	13.39	220.34
	80	59.46	282.61
RCF-CNF^2^	40	1.59	161.68
	60	7.117	195.78
	80	25.21	221.03
RCF-SiO_2_^2^	40	5.27	207.32
	60	25.41	237.24
	80	102.78	282.66
RCF-CNF^1^-SiO_2_^1^	40	4.50	191.22
	60	22.62	221.78
	80	90.28	283.62
RCF-CNF^2^-SiO_2_^2^	40	5.67	207.325
	60	25.25	251.22
	80	100.79	324.53

## Data Availability

All date used during the study appear in the submitted article.

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
