# Peer review of "High-Strength Regenerated Cellulose Fiber Reinforced with Cellulose Nanofibril and Nanosilica"

_nanomaterials, 2021, doi:10.3390/nano11102664_

Round 1
Reviewer 1 Report
The main purpose of the research should be vividly presented in Introduction. What application of the developed fiber? Synthesis and preparation method are not new as postulated in chapter 2.
3.1
The reason of the fiber strength improvement must be described with theoretical basis and scientific background. 180; Small molecules fill the gaps...is not good enough for this journal.
206; typo
Table 2; authours should explain how the crystallinity index of pulf could be evaluated.
Fig 3; five samples appear nearly same..as 256.. the intensities look almost similar considering the error of the device and measurement.
Table 4; do these elemental compositions indicate any difference?
Authors should claim this manuscript contains a topic suittable for the journal.
Reviewer 2 Report
This paper describes to prepare the cellulose composite with cellulose nanofibril and it showed the result to enhance mechanical properties due to reinforcement of nanofibril. The composites were conducted and characterized by X-ray diffraction (XRD), Fourier transform infrared spectroscopy (FT-IR) and scanning electron microscopy (SEM), and discussed about the chemical structure, crystallinity of the fibers. And also they discussed about thermal stability through thermal analysis and rheological measurement. It is very interesting concept and results, but I recommend to modify some part for publishment. 1) The preparation of composite with nanofibril was carried out with ionic liquid AMIMCl as a solvent. However, the authors didn’t mention about how much amounts it they used. They remove the ionic liquid from composite into coagulation bath. How is its quantity of AMIMCl before and after removing? 2) The authors shall check the line 102-103 about RCF-CNF1-SiO22(?), RCF-CNF1(?)-SiO22. 3) They said that CNF was dissolved in AMIMCl and regenerated into cellulose II. However, if both materials solved in AMIMCl, both materials should be homogeneous mixture. How did you explain the hypothesis for enhancement of mechanical performance? 4) The authors discussed the regeneration process to make close structure bonding from SEM images for the reason of the increased strength. However, it may be difficult to say the structure bonding from morphology in SEM images. 5) line 168: 2%wt. change to 2wt % 6) How did you prepare the samples for measurement for XRD? Powder or film? 7) The author shall describe experimental condition in detail. 8) How is the dispersibility of Si O2 in the sample? I suppose to be able observe EDX mapping image by SEM.Author Response
Please see the attachment

Round 2
Reviewer 1 Report
Prominent English typos are corrected. (eg. 193...224...)
Titles of tables and figures must be corrected too.
The reviewer can not figure out Figure 4. Re-explain and re-describe the distracting contents logically.
